# Parallel Activin and BMP signaling coordinates R7/R8 photoreceptor subtype pairing in the stochastic *Drosophila* retina

Brent S Wells, Daniela Pistillo, Erin Barnhart, Claude Desplan*

Center for Developmental Genetics, Department of Biology, New York University, New York, United States

**Abstract** *Drosophila* color vision is achieved by comparing outputs from two types of color-sensitive photoreceptors, R7 and R8. Ommatidia (unit eyes) are classified into two subtypes, known as 'pale' or 'yellow', depending on Rhodopsin expression in R7 and R8. Subtype specification is controlled by a stochastic decision in R7 and instructed to the underlying R8. We find that the Activin receptor Baboon is required in R8 to receive non-redundant signaling from the three Activin ligands, activating the transcription factor dSmad2. Concomitantly, two BMP ligands activate their receptor, Thickveins, and the transcriptional effector, Mad. The Amon TGFβ processing factor appears to regulate components of the TGFβ pathway specifically in pale R7. Mad and dSmad2 cooperate to modulate the Hippo pathway kinase Warts and the growth regulator Melted; two opposing factors of a bi-stable loop regulating R8 Rhodopsin expression. Therefore, TGFβ and growth pathways interact in postmitotic cells to precisely coordinate cell-specific output.
DOI: https://doi.org/10.7554/eLife.25301.001

## Introduction

During development, cells make both autonomous and non-autonomous decisions to give rise to functional organs. The compound eye of *Drosophila* provides a beautiful example of how these processes are integrated. It is composed of approximately 800 units, the ommatidia, each consisting of eight photoreceptor cells (R1 to R8), as well as accessory cone and pigment cells (for review see [*Hafen, 1991*]). Photoreceptors express one of six different types of photosensitive Rhodopsins (Rh). The six outer photoreceptors R1-R6 all express Rh1 and are involved in motion detection and image formation in dim light (*Heisenberg and Buchner, 1977*). In function, they resemble vertebrate rods. Their six rhabdomeres, which are made up of widely expanded membranes that contain the light-sensitive Rhs, are arranged in a trapezoid. They surround the rhabdomeres of the two inner photoreceptors R7 and R8, which are located on top of one another, thus sharing the same optic path (*Figure 1A*). R7 and R8 are involved in color vision and can be considered the equivalent of the vertebrate cones. Two primary ommatidial subtypes specialized in color vision can be identified in the main part of the retina, based on their Rh content in R7 and R8. They coordinately express UV-sensitive Rh3 in R7 with blue-Rh5 in R8 (pale ommatidia), or UV-Rh4 in R7 with green-Rh6 in R8 (yellow ommatidia). These two subtypes are stochastically distributed with a conserved ratio of 35% pale and 65% yellow (*Figure 1B*) (for review see [*Rister et al., 2013*]). Tight coupling of Rh expression in R7 with R8 likely allows flies to distinguish colors by comparing outputs from R7 and R8 cells belonging to the same ommatidium. Comparison between pale and yellow ommatidia also likely occurs, as several classes of neurons contact axons from both subtypes in the medulla (*Takemura et al., 2013*).

*For correspondence:
cd38@nyu.edu

Competing interests: The authors declare that no competing interests exist.

**Figure 1.** Ommatidia architecture and the mechanisms of TGFβ pathway signaling. (**A**) The six outer photoreceptors R1-R6 all express Rh1 and are involved in motion detection and image formation in dim light. They surround the rhabdomeres of the two inner photoreceptors R7 and R8, which are located on top of each other, thus sharing the same optic path. (**B**) The transcription factor Ss is expressed in 65% of R7 cells (**yellow** cells), activating Rh4 and repressing Rh3. Ss inhibits the instructive signal from R7 to R8, allowing for the default phosphorylation and activation of Wts in R8 and subsequent expression of Rh6 and repression of Rh5. Wts makes up one half of a double-negative feedback loop that establishes and maintains R8 subtypes. The other half of the loop, Melt, is expressed in R8s downstream of the other 35% of R7 cells (**pale** cells). These pale R7 cells lack Ss expression, allowing for expression of Rh3 and instructive signaling to R8 (red arrow). The instructive signal likely activates Melt, which represses Wts, allowing for Rh5 expression. (**C**) TGFβ superfamily ligands induce oligomerization of Type I and Type II serine-threonine kinase receptors. Binding of the ligand dimer to the Type II receptor initiates its kinase activity, phosphorylating residues on the Type I receptor, which becomes activated. Type I receptors then phosphorylate members of the receptor regulated (R)-SMAD family of transcription factors, allowing them to bind co-SMADs, translocate to the nucleus and activate or repress transcription of downstream target genes. (**D**) The TGFβ pathway in Drosophila contains both BMP and Activin subfamilies. The BMP subfamily is composed of three ligands, Dpp, Gbb and Scw, two Type I receptors, Tkv and Sax and one R-SMAD, Mad. The Activin subfamily also contains three ligands, dActβ, Daw and Myo, but only one Type I receptor, Babo and one R-SMAD, dSmad2. Both subfamilies share the Type II receptors Punt and Wit as well as the Co-SMAD, Med.
DOI: https://doi.org/10.7554/eLife.25301.002

The decision to adopt the pale or the yellow subtype is originally made in R7: In the absence of R7 cells (i.e. in a *sevenless* (*sev*) mutant), most R8 cells express Rh6, suggesting the loss of an Rh5 signal from 35% of R7 cells. In contrast, R7 cells still express Rh3 or Rh4 in the correct 35%:65% ratio when R8s are absent (*Chou et al., 1996*; *Chou et al., 1999*; *Papatsenko et al., 1997*). Therefore, a stepwise mechanism appears to be responsible for the specific expression of the different Rhs in R7 and R8: Stochastic expression of the transcription factor Spineless (Ss) in about 65% of R7 cells induces the yellow R7 subtype and represses the pale R7 subtype (*Wernet et al., 2006*). Pale R7 cells that do not express Ss activate Rh3 and signal to the underlying R8 to induce the pale subtype (Rh5). Signaling affects a bi-stable loop between the gene encoding the *Drosophila* homologue of the human LATS tumor suppressor, Warts (Wts), which is expressed in Rh6-positive yellow R8 cells, and the cell growth regulator Melted (Melt), which is co-expressed in pale R8 cells with Rh5, ensuring that the decision communicated by R7 to R8 is unequivocally fixed in R8 (*Figure 1B*) (*Mikeladze-Dvali et al., 2005*). The molecular nature of the signal from R7 to R8 has so far remained elusive. Mutations in this signaling pathway are expected to modify the expression of R8 Rhs (Rh5 and Rh6) but not R7 Rhs (Rh3 and Rh4).

We report here that signaling from R7 to R8 is mediated by both arms of the TGFβ superfamily (*Figure 1C*). In *Drosophila,* each arm (BMP and Activin) utilizes different ligands, Type I receptors and downstream R-SMADs (reviewed in [*Feng and Derynck, 2005*]): BMP homologues Decapentaplegic (Dpp), Glass Bottom Boat (Gbb), and Screw (Scw) signal through Type I receptors Thickveins (Tkv) and Saxophone (Sax) to induce phosphorylation of Mad. In the Activin branch, Activinβ (dActβ), the Activin-like protein Dawdle (Daw), and Myoglianin (Myo) signal through Type I receptor Baboon (Babo) to induce phosphorylation of dSmad2. The *Drosophila* Smad4 homologue, Medea (Med), serves as the co-SMAD for both branches of the pathway. Type II receptors Punt (Put) and Wishful Thinking (Wit) can bind to both BMP and Activin Type I receptors (*Figure 1D*). Little is known about an additional orphan ligand, Maverick (Mav), which has not been assigned to either the Activin or BMP pathway (for review see [*Peterson and O'Connor, 2014*]).

We find that both arms of the TFGβ signaling pathways specifically instruct the pale photoreceptor subset in R8. The Activin arm utilizes the three ligands dActβ, Daw, and Myo non-redundantly to activate Babo and downstream dSmad2 in R8, while the BMP arm signals by way of Dpp and Gbb to Tkv and Mad. Removing either *babo* or *tkv* leads to a dramatic increase in the yellow R8 subtype, without affecting R7 subtype. This is also true for removing pathway ligands and R-SMADs. Moreover, overexpression of constitutively activated forms of Babo or Tkv gives the opposite phenotype of high pale R8 subtypes. We show that most of the ligands are expressed in all R7 cells while regulation of their processing by the processing factor Amon appears to control their pale-specific function. We also demonstrate that both Activin and BMP signaling determine the R8 subtype by regulating the Wts/Melt bi-stable loop in pale R8 cells, revealing the presence of a crosstalk between the Hippo/Wts tumor suppressor and TGFβ pathways.

## Results

### The Type I receptors Babo and Tkv and their multiple TGFβ ligands are required to specify pale R8 subtype

To reveal factors involved in the specification of R8 subtypes, we performed an RNAi screen of transcription factors (described previously [*Hsiao et al., 2013*]) and membrane-associated factors for severe deviations from the wild type (WT) 35/65 ratio of pale/yellow ommatidia (*Figure 2A,E*). We uncovered the Activin Type I receptor Babo, which when removed with RNAi, dramatically reduced the percentage of pale R8 cells from 38% in controls to 5%. This residual Rh5 expression is very similar to the low pale R8 phenotype seen in the absence of R7 in *sev* mutants (*Chou et al., 1996*; *Papatsenko et al., 1997*) (*Figure 2B,E*, *Figure 2—figure supplement 1A*) and suggests that Babo is necessary for pale R8 subtypes. To determine whether Babo was also sufficient to drive the pale R8 subtype, we used the lGMR-Gal4 driver (*Wernet et al., 2006*) to overexpress an activated form of Babo (Babo*) in all photoreceptors R1-8. This increased pale R8 subtypes to 80% (*Figure 2C,E*). To ensure that Babo's role on R8 subtypes was not a secondary consequence of changing the R7 subtype, we examined Rh expression in R7 cells in retinas where we expressed Babo*. The ratio of Rh3/Rh4 expression in R7 was normal (35%/65%), leading to Rh4/Rh5 mispairing between R7 and R8 (*Figure 2D*), which is never observed in WT retinas (*Chou et al., 1996*) (*Figure 2—figure supplement 1B*).

Babo is the sole Type I receptor for the Activin family of ligands: dActβ, Daw, and Myo (*Figure 1D*) (*Brummel et al., 1999*). To determine whether these Activin ligands were involved in specifying R8 subtype, we removed each using the lGMR driver to express RNAi. To our surprise, we found an independent requirement for all three (*Figure 2F–I*). Testing additional RNAi constructs with different target regions for these same genes yielded similar results (*Figure 2—figure supplement 2A*).

The *babo* gene produces three distinct isoforms that vary in their ligand-binding domains: Babo-a, -b, and -c. Babo-a and Babo-b have redundant activity in specific developmental contexts (*Ng, 2008*; *Zheng et al., 2006*), although only Babo-a has been shown to specifically bind both Myo (*Awasaki et al., 2011*) and dActβ (*Song et al., 2017*). Babo-c has been shown to bind Daw (*Jensen et al., 2009*). To investigate whether one or more isoforms are involved in specifying pale subtype, we drove isoform-specific RNAi and overexpression lines using lGMR-Gal4. Removing Babo-a eliminated all pale fate (0%) while overexpressing it significantly increased pale fate (64%).

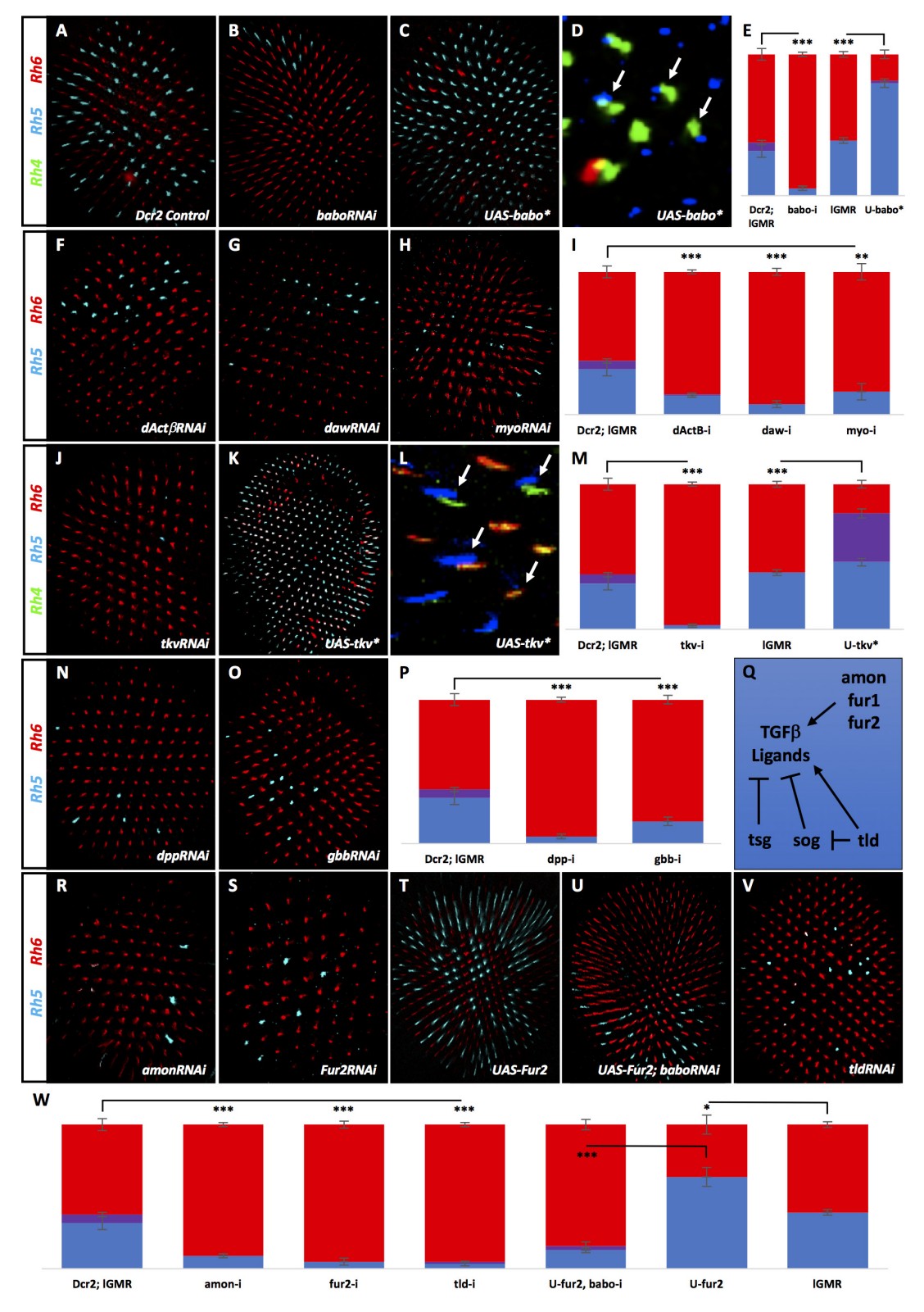

**Figure 2.** Many TGFβ ligands and receptors are required non-redundantly to specify R8 subtypes. Graph Legend: Red is the percentage of R8 photoreceptors that expressed Rh6, Blue is the percentage that expressed Rh5 and Purple is Co-expression of Rh5 + Rh6. Error bars report standard error of the mean. Paired t-tests were performed against percentages of Rh5 expression and significance is denoted where applicable (***=p≤0.01, **=p≤0.05, *=p≤0.1, ns = not significant). Where significance (or non-significance) is noted, follow the horizontal bar above the significance value to its

*Figure 2 continued on next page*

*Figure 2 continued*

termination in a vertical tick to find the comparative genotype. Percentage values for each genotype are listed as (Rh5/(Rh5 + Rh6)/Rh6). (**A**) lGMR-Gal4 > UAS-Dcr2 control retinas (32/6/62) (n = 6). Antibodies label Rh5 in blue and Rh6 in red. (**B**) lGMR-Gal4 > UAS-Dcr2; UAS-BaboRNAi drastically reduced pale subtypes (5/0/95) (n = 13). (**C**) lGMR-Gal4 > UAS-Babo* increased pale subtypes (80/2/19) (n = 20). (**D**) lGMR-Gal4 > UAS-Babo* retinas altered subtype ratios of R8 cells without altering R7, resulting in mispairing of yellow, Rh4-expressing R7 cells (green) and pale, Rh5-expressing R8 cells (blue) (arrows). (**E**) Quantification of experiments A-C with the addition of a lGMR-Gal4 control (39/0/61) (n = 7). (**F**) Using lGMR-Gal4 to express UAS-Dcr2 and UAS-dActβRNAi (14/0/86) (n = 5), UAS-DawRNAi (7/0/93) (n = 10) (**G**), and UAS-MyoRNAi (16/0/84) (n = 5) (**H**) decreased pale subtypes. (**I**) Quantification for F-H. (**J**) lGMR > UAS-Dcr2; UAS-TkvRNAi (3/0/97) (n = 6) also reduced pale R8 subtypes. (**K**) lGMR-Gal4 >UAS-Tkv* increased pale subtypes (46/34/20) (n = 16). (**L**) lGMR-Gal4 >UAS-Tkv* retinas altered the subtype ratios of R8 cells without altering R7, resulting in mispairing of yellow, Rh4-expressing R7 cells and pale, Rh5-expressing R8 cells (arrows). Some Rh5 +Rh6 co-expression occurred in UAS-Tkv* retinas (red + blue staining). (**M**) Quantification of J-K. (**N**) lGMR-Gal4 > UAS-Dcr2 and UAS-DppRNAi (5/0/95) (n = 5) or UAS-GbbRNAi (15/0/85) (n = 9) (**O**) reduced pale subtypes. (**P**) Quantification of N-O. (**Q**) Schematic of TGFβ ligand regulation: Amon, Fur1 and Fur2 are furin-type proprotein convertases capable of cleaving and activating ligands. Tsg and Sog bind ligands and prevents their activity while Tld can cleave Sog to undue repression or cleave ligands directly to activate them. (**R**) lGMR-Gal4 > UAS-Dcr2 with AmonRNAi (9/0/91) (n = 4) or Fur2RNAi (5/0/95) (n = 4) (**S**) reduced pale subtypes. (**T**) lGMR-Gal4 > UAS-Fur2 increased pale subtypes (64/0/36) (n = 3). (**U**) Removing *babo* prevented UAS-Fur2 from increasing pale subtypes (lGMR-Gal4 > UAS-Dcr2; UAS-BaboRNAi; UAS-Fur2) (13/3/84) (n = 6). (**V**) lGMR-Gal4 > UAS-Dcr2; UAS-TldRNAi (5/0/95) (n = 3) reduced pale subtypes. (**W**) Quantification of R-V.

DOI: https://doi.org/10.7554/eLife.25301.003

The following figure supplements are available for figure 2:

**Figure supplement 1.** Babo-a, -b, and Put but not Babo-c or Wit are required for R8 subtypes.

DOI: https://doi.org/10.7554/eLife.25301.004

**Figure supplement 2.** Result validation using additional alleles.

DOI: https://doi.org/10.7554/eLife.25301.005

Likewise, removing or overexpressing Babo-b significantly reduced (16%) and increased (65%) pale fate, respectively. Neither removal nor overexpression of Babo-c affected pale fate (*Figure 2—figure supplement 1C–I*). These results suggest that Babo-a and -b are each required non-redundantly for pale fate. However, Babo-a appeared to be critical for all pale fates, while removing Babo-b was far less effective. It is possible that RNAi for Babo-b might be weak and insufficient in fully knocking down endogenous activity. Indeed, overexpressing Babo-b increased pale fates similarly to Babo-a. The potential implications of these results on ligand usage are described below.

We also examined the role of Type II-receptor-binding partners, Put and Wit. We used RNAi to remove them under the control of lGMR-Gal4. We could show that Put but not Wit is required for R8 subtypes (*Figure 2—figure supplement 1A*). This is not unexpected as Put but not Wit has been described as the binding partner for Babo (*Jensen et al., 2009*). Similar to Babo, the BMP Type I receptor Tkv is also a binding partner of Put (*Ruberte et al., 1995*). To examine the role of Tkv in determining R8 subtype, we expressed TkvRNAi using lGMR-Gal4, and observed a dramatic reduction in pale R8 photoreceptors to 3%. However, removal (or overexpression) of the other BMP Type I receptor, *sax*, had no effect (*Figure 2J,M*; *Figure 2—figure supplement 2B*). To determine whether Tkv was also sufficient to induce pale R8 subtypes, we overexpressed an activated form (Tkv*) in all photoreceptors, which increased Rh5 expression to 80% (*Figure 2K,M*). Similar to Babo, manipulation of Tkv altered R8 subtype but not the R7 subtypes (*Figure 2L*). It should be noted that expressing Tkv* also resulted in co-expression of Rh5 and Rh6 in 36% of R8 cells. This appears to be due to derepression of Rh5 into yellow R8 photoreceptors since the percentage of R8 photoreceptors expressing Rh5 nearly doubled while the percentage of yellow R8s remained similar to controls.

We next tested the three BMP pathway ligands. We found roles for both Dpp and Gbb but not Scw in the specification of R8 subtype (*Figure 2N–P*, *Figure 2—figure supplement 2A*). We did not find a role for the TGFβ orphan ligand Mav (*Figure 2—figure supplement 2A*).

Maturation of TGFβ ligands requires cleavage of their precursor proteins by furin-type proprotein convertases (*Cui et al., 2001*; *Degnin et al., 2004*). In *Drosophila*, Furin 1 (Fur1) and Furin 2 (Fur2) have been shown to cleave redundantly at FSII and FSIII sites. However, Fur2 is unique in also cleaving at FSI (*Fritsch et al., 2012*; *Künnapuu et al., 2009*; *Künnapuu et al., 2014*). *Drosophila* also has a third Furin-type protease family member, Amontillado (Amon) (*Rayburn et al., 2003*) (*Figure 2Q*). As an alternative approach to verify the role of TGFβ ligands in R8 subtype specification, we knocked down factors required for their processing. Using RNAi, we found roles for Amon and Fur2 (*Figure 2R–S,W*, *Figure 2—figure supplement 2C*). Furthermore, overexpressing Fur2 in all photoreceptors increased pale R8 subtypes to 64%, an effect that was eliminated by concomitantly

removing *babo* using RNAi (13% Rh5) (*Figure 2T–U,W*). This is consistent with Babo acting downstream of Fur2-processed Activin ligands in specifying pale R8. We did not find a role for Fur1, which is expected as evidence suggests Fur2 is sufficient for cleavage at all FS sites (*Figure 2—figure supplement 2C*). The fact that Amon was also required suggests that it is not redundant with Fur2 and that each may be responsible for processing distinct TGFβ ligands (*Künnapuu et al., 2009*).

In addition to Furin-type convertases, the Tolloid (Tld) metalloprotease is also required for TGFβ signaling. Tld promotes signaling by cleaving Short Gastrulation (Sog), which dimerizes with Twisted Gastrulation (Tsg) to bind to and inhibit mature BMP ligands in the intercellular space. Cleavage of Sog disrupts the Sog/Tsg complex, releasing the ligand, which is then free to activate receptors (*Decotto and Ferguson, 2001*; *Shimmi and O'Connor, 2003*). Tld can also cleave the pro-domains of Activin ligands leading to enhanced signaling activity in the case of Daw (*Figure 2Q*) (*Serpe and O'Connor, 2006*). Consistently, Tld was required to generate the pale R8 subtype (*Figure 2V–W*, *Figure 2—figure supplement 2C*). However, removing *tsg* or *sog* did not increase pale R8 subtypes by allowing unrestricted ligand-receptor binding, nor did removing them in combination (*Figure 2— figure supplement 2C*). This may be due to limited ligands or pro-processing factors, or perhaps because different regulatory mechanisms are deployed tissue-specifically.

Collectively, the loss-of-function phenotypes for Activin (Babo) and BMP (Tkv) receptors, ligands, and ligand-processing factors strongly supports a role for the TGFβ pathway in regulating R8 cell subtype.

## Babo and Tkv are activated by canonical family ligands

Non-canonical ligand-receptor cross-activation within the TGFβ superfamily is not common but has been reported (*Peterson and O'Connor, 2014*; *Hevia and de Celis, 2013*; *Peterson et al., 2012*). Considering the involvement of ligands and receptors from both BMP and Activin families, we asked whether BMP ligands were specifically signaling through Tkv while Activin ligands were signaling through Babo. To address this, we removed specific ligands while overexpressing activated forms of Babo* or Tkv*. If Babo is downstream of Activin ligands, overexpressing Babo* should significantly rescue the loss in pale subtypes seen when removing *dActβ*, *daw*, or *myo*. Indeed, we found that this was the case (*Figure 3A–D*). However, the increase of pale/yellow R8 ratio was not as complete as in UAS-Babo* alone and was quite mild in the case of rescuing dawRNAi. We then overexpressed Tkv* while removing each of the Activin ligands individually and found that Tkv* was unable to rescue the decreased pale R8 subtypes induced by dActβRNAi and MyoRNAi. Tkv* did, however, induce enough co-expression in dawRNAi retinas to achieve a statistically significant rescue (28% Rh5) (*Figure 3E–H*). This may suggest that RNAi against daw is weakened in the UAS-Tkv* genetic background.

Likewise, we asked whether Tkv* or Babo* could rescue loss of *dpp* or *gbb*. Overexpression of Tkv* in the absence of *dpp* or *gbb* resulted in significant rescue of lost pale subtypes. However, it did not result in the same high pale/yellow ratios seen in UAS-Tkv* retinas. Similar results were obtained when expressing Babo* with dppRNAi or with gbbRNAi (*Figure 3I–L,N*). We suspect that even though Babo is not the canonical receptor for Dpp or Gbb, the activated form is capable of rescue due to Tkv-independent activation of Mad, which has been described before (*Hevia and de Celis, 2013*; *Peterson et al., 2012*) and is further addressed below.

Together, these results confirm that, for specification of the pale R8 subtype, Tkv signaling is downstream of Dpp and Gbb while Babo lies downstream of dActβ, Daw and Myo (Figure 6F).

Why is there a requirement for so many ligands and why can't Babo* or Tkv* completely rescue their loss? TGFβ ligands can heterodimerize to bind Type I receptors at the cell membrane (*Peterson and O'Connor, 2014*; *Matsuda and Shimmi, 2012*). Therefore, Dpp and Gbb may form a heteromeric complex to activate Tkv for pale R8 subtypes (*Figure 3M*). For Babo activation, dActβ, Daw, and Myo would need to form at least two dimers. A trimer would represent the first example of such a complex in TGFβ signaling.

Therefore, activating Babo in R8 might require tightly regulated levels of heterodimeric Activin ligand signaling. To explore this, we asked whether removing combinations of Activin ligands could further reduce pale subtypes in retinas. Removing either *dActβ* or *myo* in combination with *daw* did not exacerbate the decreased pale phenotype, but they did when combined with each other (1% Rh5) (*Figure 3O*). Taken together, we suggest that Babo is activated in the retina by the combined signaling of dActβ/Daw *and* Myo/Daw dimers (*Figure 3M*). This is based on the following evidence:

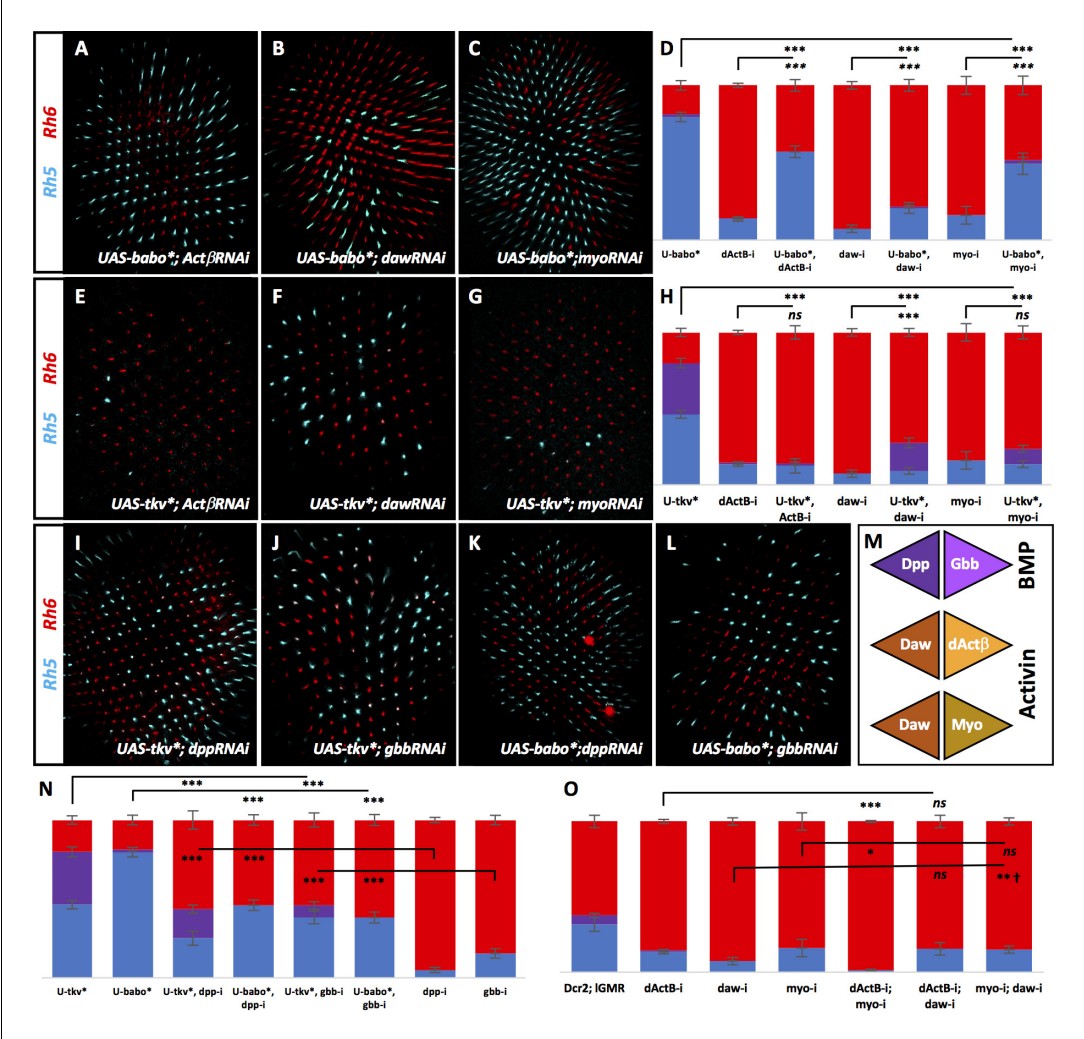

**Figure 3.** BMP and Activin ligands signal to their canonical receptors. Graph Legend: Red is the percentage of R8 photoreceptors that expressed Rh6, Blue is the percentage that expressed Rh5 and Purple is Co-expression of Rh5 + Rh6. Error bars report standard error of the mean. Paired t-tests were performed against percentages of Rh5 expression and significance is denoted where applicable (***=p≤0.01, **=p≤0.05, *=p≤0.1, ns = not significant). Where significance (or non-significance) is noted, follow the horizontal bar above the significance value to its termination in a vertical tick to find the comparative genotype. Percentage values for each genotype are listed as (Rh5/(Rh5 + Rh6)/Rh6). (A) Expressing UAS-Babo* rescued some yellow subtypes induced by removing *dActβ* (lGMR-Gal4 > UAS-Dcr2; UAS-Babo*; UAS-dActβRNAi) (57/0/43) (n = 6), *daw* (to a lesser extent) (lGMR-Gal4 > UAS-Dcr2; UAS-Babo*; UAS-DawRNAi) (21/1/78) (n = 23) (B) and *myo* (lGMR-Gal4 >UAS-Dcr2; UAS-Babo*; UAS-MyoRNAi) (49/2/48) (n = 6) (C). (D) Quantification of A-C. (E) Expressing UAS-Tkv* could not rescue yellow subtypes induced by removing *dActβ* (lGMR-Gal4 > UAS-Dcr2; UAS-Tkv*; UAS-dActβRNAi) (12/1/86) (n = 6). (F) Tkv* did slightly, yet significantly, rescue loss of *daw* (lGMR-Gal4 > UAS-Dcr2; UAS-Tkv*; UAS-DawRNAi) (9/18/72) (n = 13) but could not rescue loss of *myo* (lGMR-Gal4 > UAS-Dcr2; UAS-Tkv*; UAS-MyoRNAi) (14/10/76) (n = 8) (G). (H) Quantification of E-G. (I) Expressing UAS-Tkv* rescued many yellow subtypes induced by removing *dpp* (lGMR-Gal4 > UAS-Dcr2; UAS-Tkv*; UAS-DppRNAi) (25/18/56) (n = 15) or *gbb* (lGMR-Gal4 > UAS-Dcr2; UAS-Tkv*; UAS-GbbRNAi) (38/8/54) (n = 12) (J). (K) Expressing UAS-Babo* also rescued many yellow subtypes induced by removing *dpp* (lGMR-Gal4 > UAS-Dcr2; UAS-Babo*; UAS-DppRNAi) (46/0/54) (n = 7) or *gbb* (lGMR-Gal4 > UAS-Dcr2; UAS-Babo*; UAS-GbbRNAi) (38/0/62) (n = 6) (L). (M) Schematic for hypothetical ligand dimerization possibilities upstream of Tkv and Babo activation. Dpp + Gbb may heterodimerize to activate Tkv while Daw + dActβ and Daw + Myo may heterodimerize and signal at highly regulated levels for Babo activation. (N) Quantification of I-L. (O) Co-expression of dActβRNAi and MyoRNAi (lGMR-Gal4 > UAS-Dcr2; UAS-dActβRNAi; UAS-MyoRNAi) (1/0/99) (n = 6) exacerbated specification of yellow subtypes induced in dActβRNAi or MyoRNAi retinas alone. However, combining DawRNAi with either dActβRNAi (lGMR-Gal4 > UAS-Dcr2; UAS-dActβRNAi; UAS-DawRNAi) (15/0/85) (n = 7) or MyoRNAi (lGMR-Gal4 > UAS-Dcr2; UAS-MyoRNAi; UAS-DawRNAi) (15/0/85) (n = 12) could not exaggerate the phenotypes of any of the three alone. † denotes a change in Rh5/Rh6 ratio that was significant but in the unexpected direction.

DOI: https://doi.org/10.7554/eLife.25301.006

(1) Removing *daw* resulted in a higher percentage of yellow R8 cells than when either *dActβ* or *myo* were removed, presumably because it disrupts both potential heterodimers (*Figure 3O*). (2) Removing *daw* with either *dActβ* or *myo* did not exaggerate the *daw* phenotype while removing *dActβ +myo* (which would affect both heterodimers) resulted in exaggerated yellow R8 subtypes relative to either *dActβ* or *myo* alone (*Figure 3O*). (3) Babo-a and Babo-b isoforms were required for pale subtypes while Babo-c is dispensable (*Figure 2—figure supplement 1C–I*). Since Daw normally utilizes the Babo-c isoform for signaling (*Jensen et al., 2009*), heterodimerization with dActβ or Myo may be required to enable its signaling in pale subtype specification.

## Babo instructs R8 subtype through dSmad2 while Tkv does so through Mad

Mad represents the R-SMAD downstream of Tkv activity while dSmad2 acts downstream of Babo (*Figure 1D*) (*Peterson and O'Connor, 2014*). Consistent with its canonical role in Babo signaling, removing *dSmad2* resulted in a sharp decrease in pale R8 subtypes (15%) (*Figure 4A,E*), a decrease that was still observed even when Babo* was overexpressed (23%) (Babo*+dSmad2 RNAi; *Figure 4B,E*). Thus, dSmad2 is required downstream of Babo signaling for specification of the pale R8 subtype. Removing *mad* with RNAi also resulted in a decrease of pale R8 subtypes (4%) (*Figure 4C,E*), even when overexpressing Tkv* (6%) (Tkv*+madRNAi; *Figure 4D–E*), placing Mad downstream of Tkv signaling in specifying pale R8 subtypes. Removing *med*, like removing its binding partners *dSmad2* or *mad*, also resulted in loss of pale R8 subtypes (15%) (*Figure 4—figure supplement 1A*).

## Babo and Tkv are required upstream of Melt, Wts and Yorkie to regulate pale R8 subtypes

Previous studies have shown that inhibitory feedback between the Hippo pathway kinase, Wts, and the PH domain protein, Melt, establish and maintain pale R8 subtypes downstream of an R7 to R8 signal (*Figure 1B*) (*Mikeladze-Dvali et al., 2005*). Wts and Melt are expressed in a mutually exclusive pattern with Wts in Rh6-expressing yellow R8 and Melt in pale R8 (Rh5) (*Figure 4F*). Removing *wts* results in 100% pale R8 subtypes while removing *melt* results in nearly 100% yellow R8 subtypes (*Figure 4G–H,T*). We therefore asked whether Melt and Wts acted downstream of Babo and Tkv signaling. Expressing Babo* in all ommatidia led to loss of Wts and subsequent expression of Melt and Rh5 in 80% of R8 cells (*Figure 4I*; *Figure 4—figure supplement 1B*). Removing *melt* while expressing Babo* resulted in yellow R8 subtypes in 100% of ommatidia (*Figure 4J,T*). Similar to Babo*, expressing Tkv* also activated Melt and Rh5 with corresponding loss of Wts. In some photoreceptors that co-expressed Rh5 +Rh6, Wts expression was reduced but not completely lost and, in these ommatidia, Melt was not induced (*Figure 4K*; *Figure 4—figure supplement 1C* – yellow arrowheads). Removing *melt* resulted in loss of Tkv*-induced pale R8 subtypes (10%) (*Figure 4L,T*). Removing *wts* increased pale R8 subtypes to almost 100% from the low levels seen when removing either *dSmad2* (WtsRNAi + dSmad2 RNAi) or *mad* (WtsRNAi + MadRNAi) (*Figure 4M–N,T*). These results suggest that Wts and Melt are epistatic to Babo and dSmad2 as well as Tkv and Mad.

The coactivator Yorkie (Yki) acts with the transcription factor Scalloped (Sd), downstream of Wts. Downregulation of Wts allows for Yki/Sd induction of Rh5 expression (*Jukam et al., 2013*). Consistent with this, we found that *yki* was indeed epistatic to both Babo/dSmad2 and Tkv/Mad for Rh5 regulation (*Figure 4O–T*).

## TGFβ ligands are processed in pale R7 cells and activate their receptors and subsequent R-SMADs in R8 cells

To determine in which cells receptors were acting to confer pale R8 subtype, we removed specific factors using cell-specific Gal4 drivers. We removed *babo*, *tkv*, *mad*, and *dSmad2*, which we expect to each act in R8 cells. Removing them in R7 (using PM181-Gal4 + Pros-Gal4) had no effect on pale R8 subtypes (*Figure 5A*), while removing them in R8 (using Sca-Gal4 + Sens-Gal4) dramatically decreased Rh5 expression (*Figure 5B*). Furthermore, the decreased number of pale R8 subtypes observed when removing either *babo* or *tkv* in all photoreceptors under the control of lGMR-Gal4 was returned to wild-type levels by co-expressing the Gal4 inhibitor Gal80 specifically in R8 (*Figure 5C*).

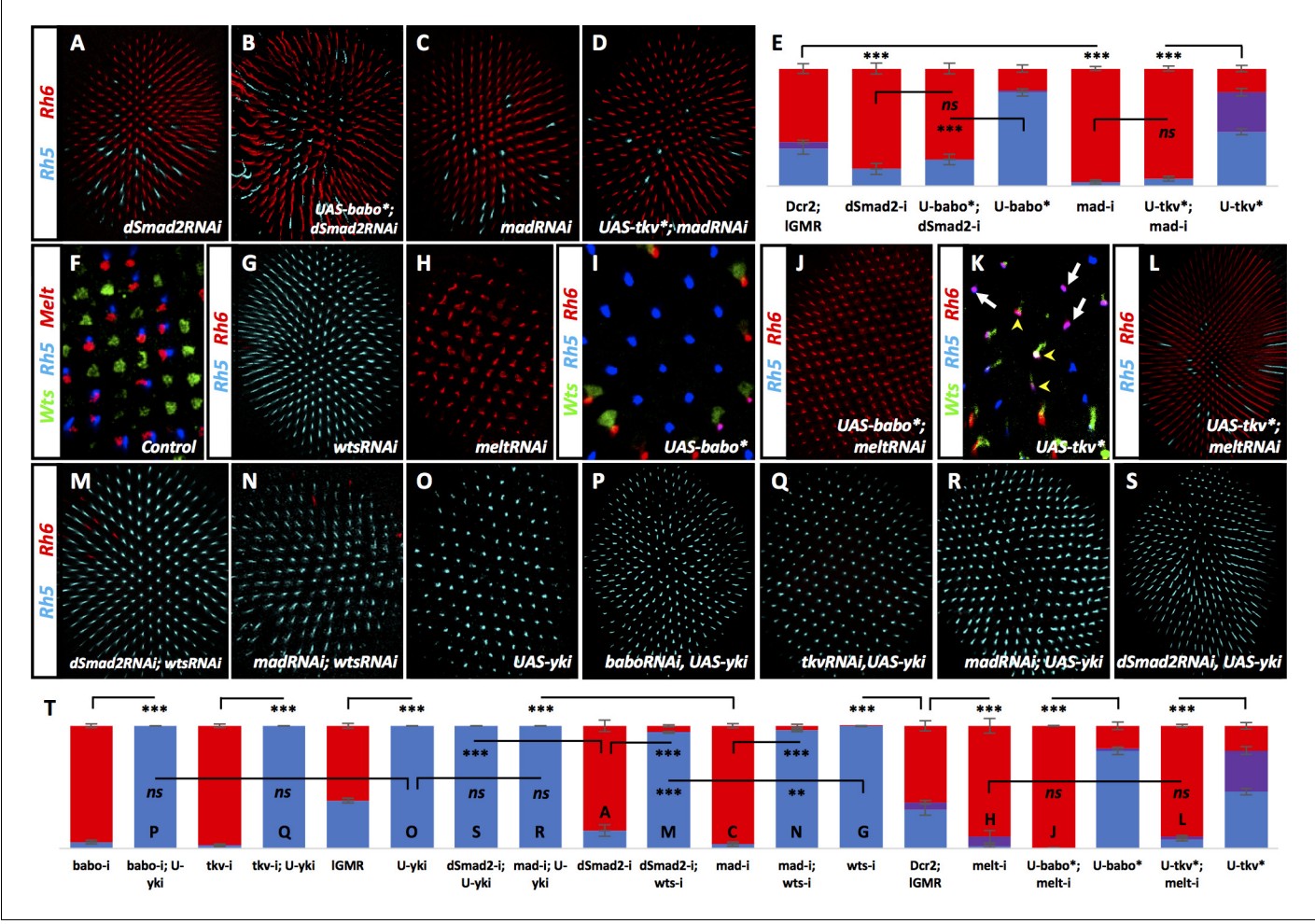

**Figure 4.** Babo, through dSmad2 and Tkv, through Mad, regulate Wts and Melt. Graph Legend: Red is the percentage of R8 photoreceptors that expressed Rh6, Blue is the percentage that expressed Rh5 and Purple is Co-expression of Rh5 +Rh6. Error bars report standard error of the mean. Paired t-tests were performed against percentages of Rh5 expression and significance is denoted where applicable (***=p ≤ 0.01, **=p ≤ 0.05, ns = not significant). Where significance (or non-significance) is noted, follow the horizontal bar above the significance value to its termination in a vertical tick to find the comparative genotype. Percentage values for each genotype are listed as (Rh5/(Rh5 +Rh6)/Rh6). (A) Expressing UAS-Dcr2 and UAS-dSmad2RNAi with lGMR-Gal4 reduced pale subtypes (lGMR-Gal4 >UAS-Dcr2; UAS-dSmad2RNAi) (15/0/85) (n = 4) and this reduction was maintained even when UAS-Babo* was co-expressed (lGMR-Gal4 >UAS-Dcr2; UAS-dSmad2RNAi; UAS-Babo*) (23/0/77) (n = 7) (B). (C) Expressing UAS-Dcr2 and UAS-madRNAi with lGMR-Gal4 also reduced pale subtypes (lGMR-Gal4 >UAS-Dcr2; UAS-MadRNAi) (4/0/96) (n = 7). (D) Expressing UAS-Tkv* could not rescue MadRNAi-induced loss of pale subtypes (lGMR-Gal4 >UAS-Dcr2; UAS-MadRNAi, UAS-Tkv*) (6/0/94) (n = 9). (E) Quantification of A-D. (F) Wild-type retinas express Melt (red) and Wts (green) in mutually exclusive subsets of R8 cells; Melt (red) with Rh5 (blue) in the pale subset and Wts (green) with Rh6 (not shown) in the alternate, yellow R8 subset. (G) Removing *wts* (lGMR-Gal4 > UAS-Dcr2; UAS-WtsRNAi) (100/0/0) (n = 8) led to total loss of yellow subtype while removing *melt* (lGMR-Gal4 > UAS-Dcr2; UAS-MeltRNAi) (2/8/90) (n = 5) resulted in the opposite, loss of pale subtypes (H). (I) Expressing Babo* increased pale subtypes (blue), which was concomitant with a loss of Wts expression (green). (J) Removing *melt* in retinas that overexpress Babo* (lGMR-Gal4 > UAS-Dcr2; UAS-Babo*; UAS-MeltRNAi) (0/0/100) (n = 11) prevented Babo*-induced pale subtypes. (K) Overexpression of Tkv* increased the number of ommatidia expressing Rh5 (blue) in R8. In this genotype, increased Rh5 expression was often co-expressed with Rh6 (magenta). Increased Rh5 expression was coincident with loss of Wts, even in some instances where Rh6 was still present (white arrows); however, Wts remained in other R8 cells that co-expressed Rh5 + Rh6 (yellow arrowheads). (L) Removing *melt* from Tkv* retinas (lGMR-Gal4 > UAS-Dcr2; UAS-Tkv*; UAS-MeltRNAi) (7/3/90) (n = 46) resulted in a loss of Tkv*-induced Rh5 expression. (M) While significantly different from wtsRNAi controls (due to extremely low variance), removing *wts* prevented increased yellow subtypes seen in dSmad2RNAi (lGMR-Gal4 > UAS-Dcr2; UAS-dSmad2RNAi; UAS-WtsRNAi) (95/0/5) (n = 6) and MadRNAi (lGMR-Gal4 > UAS-Dcr2; UAS-MadRNAi; UAS-WtsRNAi) (97/0/3) (n = 12) (N) retinas. (O) lGMR-Gal4 > UAS Yki retinas exhibited 100% pale subtypes (100/0/0) (n = 7). (P) Expressing UAS-Yki overcame the high incident of yellow subtypes induced by expressing BaboRNAi (lGMR-Gal4 > UAS-Dcr2; UAS-BaboRNAi; UAS-Yki) (100/0/0) (n = 6), TkvRNAi (lGMR-Gal4 > UAS-Dcr2; UAS-TkvRNAi; UAS-Yki) (100/0/0) (n = 6), MadRNAi (lGMR-Gal4 >UAS-Dcr2; UAS-MadRNAi; UAS-Yki) (100/0/0) (n = 6) (R) and dSmad2RNAi (lGMR-Gal4 >UAS-Dcr2; UAS-dSmad2RNAi; UAS-Yki) (100/0/0) (n = 6) (S). (T) Quantification of G-H, J, L-S. Letters (P, Q, O. . .) have been added to applicable bars to help identify corresponding image panels.

*Figure 4 continued on next page*

*Figure 4 continued*

DOI: https://doi.org/10.7554/eLife.25301.007

The following figure supplement is available for figure 4:

**Figure supplement 1.** Med, Babo, and Tkv are required for R8 subtypes.

DOI: https://doi.org/10.7554/eLife.25301.008

Additionally, we asked whether expressing Babo* or Tkv* could rescue the decreased pale R8 subtypes in a *sev* mutant that lacks R7 cells and – due to lack of signaling – exhibits almost entirely yellow R8 subtypes (*Figure 2—figure supplement 1A*) (*Chou et al., 1999*). We found an increase in pale subtypes similar to phenotypes observed in WT retinas, when either Babo* (*Figure 5E–F*) or Tkv* (*Figure 5E,G*) was expressed, suggesting that both can induce pale R8 subtypes even in the absence of R7 cells. Together, these results suggest that Babo, Tkv, Mad, and dSmad2 are required in R8 cells to induce the expression of Rh5.

We then asked where the signaling ligands were being expressed. We examined ligand reporter constructs between 45 and 55 hr after pupation (APF), when subtype decisions are thought to occur (*Jukam and Desplan, 2011*). A protein reporter for dActβ was strongly expressed in R7 photoreceptors, showing overlap with the transcription factor R7 marker, Prospero (Pros), as well as in some R3 cells (*Figure 5H–H"*). To further examine whether dActβ was expressed in pale or yellow R7 cells, we co-stained with Ss, which is expressed specifically in yellow R7s at this stage of their development. Here we found that dActβ was expressed in all R7 cells (*Figure 5I–I'* - Ss-positive cells circled in purple, Ss-negative cells circled in red). Since pale R8 subtype is dependent upon the signaling of multiple ligands, another ligand might be uniquely expressed in pale R7 cells. To this end, we investigated additional reporter constructs for Myo, Dpp and Daw. We found that Myo was expressed in all R4 and R7 cells (*Figure 5J–J"*) and that Dpp was expressed weakly in all photoreceptors (*Figure 5K'*, arrows). A Daw transcriptional reporter did not yield conclusive results and may be expressed at levels too low to detect or it might report on an enhancer region unused in the retina.

Given that ligand expression was promiscuous amongst ommatidia subtypes, we hypothesized that ligand activation by processing factors could provide a subtype-specific signaling mechanism. To explore this, we examined a reporter for the processing factor Amon, which, when removed, resulted in decreased pale subtypes (*Figure 2R,W*). Consistent with our hypothesis, we found that Amon was expressed almost exclusively in pale R7 cells (*Figure 5L–L"* – circled in purple) and very rarely overlapped with Ss-positive yellow R7 cells (1%) (*Figure 5L–L"* – circled in red).

However, many pale R7 cells were also not positive for Amon (*Figure 5L–L"* – circled in white). This may be due to transient expression of Amon in activating ligands to instruct pale subtype in R8 cells. Pupal retinas expressed Amon non-uniformly: Some regions expressed Amon in a percentage of R7 cells similar to what is observed for pale R7s in WT retinas and some in many fewer (*Figure 5M–M'* – compare 23% with 4% in the same retina). This could be due to temporally distinct activation of Amon within regions of the retina coupled with its transient nature. Consistent with this, some pupal retina (45–55 hr APF) exhibited high Amon expression (>30%) in pale R7s, which was completely absent in slightly older pupae. However, even across expanded developmental time, Amon expression in pale R7s averaged 9%, which is highly significant compared to 1% in yellow cells (9.9E-06 using a paired t-test). Amon was also expressed in outer photoreceptors where it may be required for processing ligands that are expressed in those cells for a different function (*Figure 5L–L"* – circled in blue). Rare expression of Amon in yellow R7 cells may represent a signaling mistake, which would result in Rh4/Rh5 mispairing in adults, although this is almost never observed. This would lend reason for non-redundant signaling by multiple TGFβ ligands; alternative regulation of other ligands would prevent the propagation of such a mistake.

We also examined a transcriptional reporter for the processing factor Fur2, which was required for pale subtypes (*Figure 2S–U,W*), but found no expression in photoreceptors. This may be due to the nature of the Gal4 insertion, which may miss an important enhancer. Taken together, these results provide a mechanism for ligand activation specific to pale ommatidia, capable of driving R8 subtype downstream of Ss-negative pale R7 cells (*Figure 6F*).

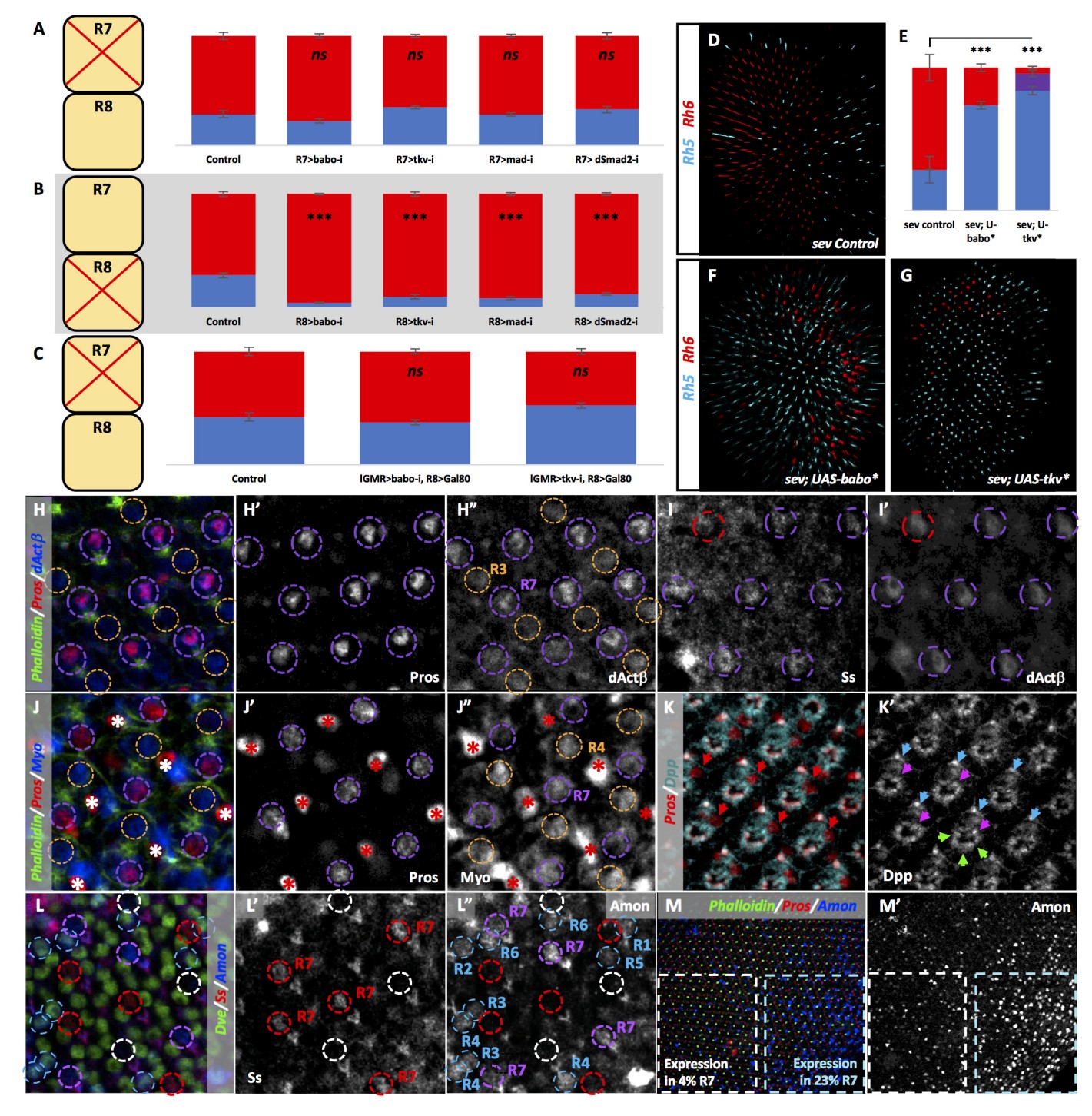

**Figure 5.** TGFβ ligands are processed specifically in pale R7 and activate their receptors and subsequent R-SMADs in R8. Graph Legend: Red is the percentage of R8 photoreceptors that expressed Rh6, Blue is the percentage that expressed Rh5 and Purple is Co-expression of Rh5 +Rh6. Error bars report standard error of the mean. Paired t-tests were performed against percentages of Rh5 expression and significance is denoted where applicable (***=p ≤ 0.01, ns = not significant). Percentage values for each genotype are listed as (Rh5/(Rh5 +Rh6)/Rh6). Control for A is PM1.81-Gal4; Pros-Gal4 >UAS-Dcr2 (29/0/71) (n = 6). For B, the control is Sca-Gal4; Sens-Gal4 > UAS-Dcr2 (28/0/72) (n = 7). Control for C is lGMR-Gal4 > UAS-Dcr2; Sca-lexA > lexAOP-Gal80 (42/0/58) (n = 7). (**A**) Removing *babo* (PM1.81-Gal4; Pros-Gal4 > UAS-Dcr2; UAS-BaboRNAi) (23/0/77) (n = 6), *tkv* (PM1.81-Gal4; Pros-Gal4 > UAS-Dcr2; UAS-TkvRNAi) (35/0/65) (n = 7), *mad* (PM1.81-Gal4; Pros-Gal4 >UAS-Dcr2; UAS-MadRNAi) (28/0/72) (n = 5), or *dSmad2* (PM1.81-Gal4; Pros-Gal4 >UAS-Dcr2; UAS-dSmad2RNAi) (33/0/67) (n = 6) specifically in R7 had no effect on Rh5/Rh6 ratios. (**B**) Removing *babo* (Sca-Gal4; Sens-Gal4 > UAS-Dcr2; UAS-BaboRNAi) (4/0/96) (n = 6), *tkv* (Sca-Gal4; Sens-Gal4 > UAS-Dcr2; UAS-TkvRNAi) (10/0/90) (n = 7), *mad* (Sca-Gal4; Sens-

*Figure 5 continued on next page*

*Figure 5 continued*

Gal4 > UAS-Dcr2; UAS-MadRNAi) (8/0/92) (n = 4), or *dSmad2* (Sca-Gal4; Sens-Gal4 > UAS-Dcr2; UAS-dSmad2RNAi) (12/0/88) (n = 4) specifically in R8 reduced the number of pale subtypes as we saw with lGMR-Gal4 drivers. (C) Silencing Gal4 expression specifically in R8 attenuated the lGMR-Gal4 > UAS BaboRNAi- and TkvRNAi-induction of yellow R8 subtypes (lGMR-Gal4 > UAS-Dcr2; UAS-BaboRNAi; Sca-lexA > lexAOP-Gal80) (37/0/63) (n = 6) (lGMR-Gal4 > UAS-Dcr2; UAS-TkvRNAi; Sca-lexA > lexAOP-Gal80) (52/0/48) (n = 4). (D) *sev* mutations genetically ablate R7 cells resulting in increased amounts of default yellow R8 cells. Our *sev* mutant control, which carries a lGMR-Gal4 driver, expressed higher levels of pale subtypes (29/0/71) than observed in *sev* mutants in a WT background (4/0/96), likely due to genetic background (n = 3). (E) Quantification of D, F-G. (F) Expressing UAS-Babo* in a *sev* mutant (*sev$^{14}$*; lGMR-Gal4 > UAS-Babo*) (74/0/26) (n = 5) increased the number of pale subtypes significantly. (G) Expressing UAS-Tkv* in a *sev* mutant (*sev$^{14}$*; lGMR-Gal4 > UAS-Tkv*) (84/12/4) (n = 5) increased pale subtypes significantly. (H-M) Pupal retinas, 45–55 hr APF. (H) dActβ was expressed in all R7 cells (circled in purple) and less strongly in some R3 cells (circled in orange). (H′) Single-channel staining of the Prospero (Pros) transcription factor, which is expressed in all R7 cells. (H′). Single-channel dActβ protein reporter overlaped with Pros expression. (I) The Spineless (Ss) antibody marks yellow R7 cells (circled in purple). (I′) dActβ was expressed in both Ss-positive yellow R7 cells (circled in purple) and Ss-negative pale R7 cells (circled in red). (J) A Myo protein reporter was expressed in all R7 cells (circled in purple) as well as R4 cells (circled in orange). Bristle cells also labeled brightly (marked with asterisks). (J′) Single-channel Pros staining. (J′) Single-channel Myo reporter. (K) A Dpp-GFP transcriptional reporter was expressed in all photoreceptors. Pros marks R7 cells (red arrows). (K′) Single-channel Dpp reporter. Blue arrows mark R6 expression; green arrows mark expression in photoreceptors 1–5; pink arrows mark expression in R7. (L) An Amon-Gal4 transcriptional reporter was expressed in some pale R7 cells (circled in purple) but is absent in others (circled in white). It was absent from yellow R7 cells (circled in red). The reporter was also expressed in some outer photoreceptors (circled in blue). (L′) Single channel Ss staining marks yellow R7 cells (circled in red). (L′) Single-channel Amon reporter. Amon pale R7 expression – 9%, yellow R7 expression – 1%. N = 2678 ommatidia. (M) Amon expression was variable across individual pupal retina. (M′) Single channel Amon reporter expression.

DOI: https://doi.org/10.7554/eLife.25301.009

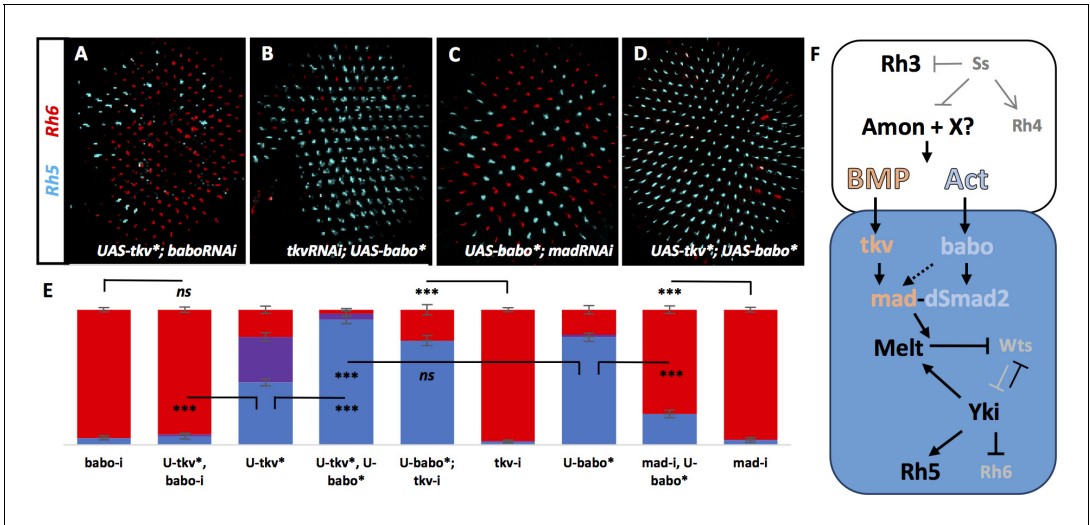

**Figure 6.** Tkv and Babo signal in parallel to establish a 'double-checked' pathway, ensuring correct R7/R8 pairing. Graph Legend: Red is the percentage of R8 photoreceptors that expressed Rh6, Blue is the percentage that expressed Rh5 and Purple is Co-expression of Rh5 + Rh6. Error bars report standard error of the mean. Paired t-tests were performed against percentages of Rh5 expression and significance is denoted where applicable (***=p ≤ 0.01, ns = not significant). Where significance (or non-significance) is noted, follow the horizontal bar above the significance value to its termination in a vertical tick to find the comparative genotype. Percentage values for each genotype are listed as (Rh5/(Rh5 + Rh6)/Rh6). (A) Removing *babo* resulted in a high incidence of yellow subtypes even when UAS-Tkv* was expressed (lGMR-Gal4 > UAS-Dcr2; UAS-BaboRNAi; UAS-Tkv*) (6/2/92) (n = 6). (B) Removing *tkv* could not eliminate increased pale subtypes in UAS-Babo* retinas (lGMR-Gal4 > UAS-Dcr2; UAS-TkvRNAi; UAS-Babo*) (77/0/23) (n = 9). (C) Removing *mad* significantly reduced pale subtypes induced by expressing UAS-Babo* (lGMR-Gal4 > UAS-Dcr2; UAS-MadRNAi; UAS-Babo*) (23/0/77) (n = 15). (D) Expressing both UAS-Tkv* and UAS-Babo* (lGMR-Gal4 > UAS-Tkv*; UAS-Babo*) (93/5/3) (n = 7) significantly increased pale subtypes above that induced by either alone. (E) Quantification of A-D. (F) Model. Lack of *ss* in pale R7 cells allows for expression of Amon and possibly other proprotein convertases to activate signaling of BMP and Activin ligands, which activate canonical receptors and R-SMADs Tkv/Mad and Babo/dSmad2, respectively, in R8. Cooperative action of Mad and dSmad2 then regulate the Melt/Wts bi-stable loop, allowing Yki/Sd to induce expression of Rh5 and repression of Rh6.

DOI: https://doi.org/10.7554/eLife.25301.010

## Tkv and Babo signaling establish a 'double-checked' pathway, ensuring correct R7/R8 pairing

Next, we examined the relationship between Tkv and Babo in specifying the pale R8 subtype. In most developmental contexts, only one TGFβ pathway – either BMP or Activin – is utilized. Why then is there a requirement for both in R8 specification? To examine this question, we asked how each pathway depended upon the other. Expression of Tkv*+BaboRNAi resulted in loss of pale subtypes similar to BaboRNAi alone (8% vs 5%) (*Figure 6A,E*), while expression of TkvRNAi + Babo* resulted in a gain of pale subtypes similar to Babo* alone (77% vs 80%) (*Figure 6B,E*). These results suggest that Babo is epistatic to Tkv, presumably: Tkv→Mad→Babo→dSmad2. If this were true, Babo* should be sufficient to rescue MadRNAi-induced yellow R8 subtypes (Babo* + MadRNAi). However, this was not the case (*Figure 6C,E*). Alternatively, Tkv might activate Babo, which then activates both Mad and dSmad2 to drive pale R8 subtypes. This would be consistent with ours and other's data showing activation of Mad by Babo (*Figure 3K,N*) (*Hevia and de Celis, 2013*; *Peterson et al., 2012*; *Bickel et al., 2008*; *Gesualdi and Haerry, 2007*). If this were true, simultaneously expressing activated forms of Tkv* and Babo* (Tkv* + Babo*) should be identical to Babo* alone. However, we observed a significant increase in pale subtypes above what was seen when either Babo* or Tkv* were expressed alone (*Figure 6D–E*). Therefore, parallel activity of both Babo/dSmad2 and Tkv/Mad appears to be required for specification of pale subtypes. Perceived epistasis between Babo and Tkv is likely due to the ability of Babo to activate some levels of Mad independent of Tkv, allowing for necessary signaling by both dSmad2 and Mad. In contrast, Tkv signaling in the absence of Babo (Tkv* + BaboRNAi) would only activate Mad and fail to specify pale subtypes without dSmad2 activity. We propose a model where pale R7-specific ligand processing by Amon and possibly additional factors instructs non-redundant signaling by two arms of the TGFβ family to specify pale R8 subtypes as part of a 'double-check' mechanism, ensuring correct R7/R8 photoreceptor pairing (*Figure 6F*).

dSmad2 and Mad may work as part of a complex (*Grönroos et al., 2012*), or independently, to bind enhancers of the *wts* or *melt* genes and regulate their expression. In this case, the stoichiometry of Babo and Tkv activity could be important. Retinas that lack both Melt and Wts exhibit the *wts* mutant phenotype, suggesting that Wts is downstream of Melt regulation (*Mikeladze-Dvali et al., 2005*). Therefore, the specification of pale subtypes may occur through positive regulation of Melt rather than negative regulation of Wts by Mad and dSmad2.

## Discussion

Here, we describe the long-sought signaling mechanisms that ensure correct pairing of yellow R7 with yellow R8, and pale R7 with pale R8 during *Drosophila* retinal development. Our results describe non-redundant, parallel signaling from both Activin and BMP pathways that converge to activate Melt, thereby inhibiting Wts and instructing the pale subtype in R8 photoreceptors. In one arm of a double-checked mechanism for subtype specification, Activin pathway ligands dActβ, Daw, and Myo induce Babo activation of dSmad2. A second arm utilizes BMP ligands Dpp and Gbb to instruct Tkv activation of Mad. Repression of Hippo signaling in R8 cells then allows for Yki to mediate, along with Sd, the expression of Rh5 and repression of Rh6 (*Figure 6F*). Our results provide the first evidence for interaction between TGFβ signaling and Melt in *Drosophila*, although it has been recently shown in mouse cell culture (*Shathasivam et al., 2015*), and therefore may represent a conserved interaction across species.

Furthermore, we find that cell fate may be regulated through pale-localized processing of widely expressed TGFβ ligands. In *Drosophila*, localized furin-type proprotein convertases have been shown to dictate gradients of BMP ligands in the developing embryo and imaginal discs, regulating critical developmental boundaries (*Künnapuu et al., 2014*; *Sopory et al., 2010*). However, to our knowledge, this would be the first example in which the specific localization of a convertase is being utilized to specify fate. Many convertases have previously been shown to be broadly expressed in specific tissue types, though some, including furin-type Class one convertases are ubiquitous and widespread (reviewed in [*Seidah et al., 1998*]). Few mechanisms for such tight regulation of convertases have been observed and include localized convertase inhibitory signals such as Emilin-1 and ESL-1, described in mice, alternative convertase motifs within the same ligand; and distinctions in convertase recognition motifs and flanking sequences (reviewed in [*Constam, 2014*]). Here, we

describe ligand regulation by a convertase on a very precise scale. This suggests that widespread transcription of ligands could be fine-tuned post-transcriptionally to specify fate in close-knit groups of cells making up complex organs. More work is necessary to determine how this process is regulated upstream of convertases and specifically how Ss might be preventing expression of Amon, and possibly Fur2, in yellow R7s.

Ligands of the TGFβ superfamily have been implicated in numerous morphogenetic developmental processes that depend on long-range diffusion to instruct regional patterning over a distance of many cell diameters (*Gurdon and Bourillot, 2001*; *Lawrence et al., 1972*; *Wolpert, 1989*). While it may not be surprising to see the reuse of patterning systems already utilized in early eye morphogenesis and axonal targeting (*Heberlein et al., 1993*; *Zheng et al., 2003*; *Ting et al., 2014*), it is unclear why or how such long-range signals are deployed for one R7 cell to instruct only the underlying R8 and how that signal is confined to a single ommatidial cartridge, and even from R7 to R8, without accidental activation from an outer photoreceptor, where several signals are expressed. Use of a long-range signaling mechanism would require insulation of ommatidial arrays and photoreceptors to prevent ligand 'leakage' to internal and surrounding R8 cells, which would result in mispairing and large, non-stochastic 'clumps' of similar subtype. Indeed, confinement of cell signaling may reflect a need for the intact boundaries observed between ommatidia throughout pupal development (reviewed in [*Treisman, 2013*]). There are many described mechanisms for limiting the diffusion of morphogens (reviewed in [*Rogers and Schier, 2011*]). One possible mechanism for limiting the action of morphogens to single-cell resolution has been described in the ovarian niche: here BMP molecules are required to specify adjacent germline stem cells (GSC). This is achieved through expression of the glypican, Dally, in cap cells, which binds and stabilizes BMPs on the surface of the contacting cell. Signaling is then mediated through physical contact of cap cell and GSC (*Guo and Wang, 2009*; *Dejima et al., 2011*). A similar mechanism may exist in inner R7 and R8 cells, which sit adjacently, to help facilitate TGFβ signaling specifically in the pale subtype.

Previous work from our lab outlined an intricate network of factors necessary for post-mitotic control of R8 subtype (*Mikeladze-Dvali et al., 2005*; *Jukam et al., 2013*; *Jukam and Desplan, 2011*). There, the Hippo tumor suppressor pathway has been repurposed to regulate R8 Rh expression. We find that TGFβ signaling can switch off the Hippo pathway to establish pale fate, which is maintained by further Wts/Melt feedback. Though there are several examples of signaling that regulate the Hippo pathway during growth (reviewed in [*Grusche et al., 2010*]), few have been shown to have a capacity for specification of cell fate. Our work suggests that many roles of Hippo signaling may remain uncovered.

Why use two signaling pathways to activate pale fate? One possibility is that combinatorial signaling results in synergistic activation of Rh5 and repression of Rh6. Indeed, activating either Babo* or Tkv* resulted in ~80% pale subtypes while Babo*+Tkv* resulted in 98%. Synergistic activation between cAMP and Arachidonic Acid signaling pathways has been described in vertebrate steroidogenesis (*Stocco et al., 2005*) and suggests that the stoichiometry of each signaling pathway is critical. Stoichiometry would ultimately impact the level of dSmad2 and Mad and may play a role in binding site affinity or in formation of a transcriptional complex. Alternatively, Rh5 expression may require a temporal series of events: Babo and dSmad2 might first inactivate Wts, followed by Tkv/Mad activation of Melt, again requiring both for robust activation. This may explain the observed Rh5 + Rh6/Wts co-expression when we drive Tkv*: If an R8 cell is first specified as yellow (lack of *babo*), subsequent activation of Tkv* could induce expression of Rh5 without completely eliminating Wts or Rh6 (see *Figure 4K*, yellow arrowheads). Authentication of a signaling event by using two pathways would also help to prevent misexpression of pale fate target genes in a yellow ommatidium, that is Rh4 in R7 coupled with Rh5 in R8, which is never observed in the WT.

In general, removing components of either Activin or BMP signaling pathways reduces the ratio of pale subtypes to levels observed when genetically ablating all R7 cells in *sev* mutants (4%) (*Figure 2—figure supplement 1A*), suggesting that they account for the entire instructive signal. But what accounts for the remaining 4% pale subtypes in *sev* mutant retinas? This could represent an ancestral pathway that allowed for an autonomous subtype choice in R8 before the divergence of pale and yellow R7 cell types. Indeed, the UV Rhs duplicated early in the dipteran lineage while R8 expressing Blue or Green Rhs are more ancient (*Pichaud et al., 1999*). Yet, local cell-cell interactions are a conserved mechanism for photoreceptor pairing across evolutionarily distant species. In the butterfly, *P. xuthus*, stochastic Ss expression directs the specification of additional types of

photoreceptors through the addition of a second R7 cell per ommatidium, thus generating a third ommatidial subtype. The choice made in each of the two R7s dictates which Rhs are expressed in surrounding photoreceptors, similarly to the way R7 dictates R8 subtype in *Drosophila*. Interestingly, photoreceptors homologous to the outer photoreceptors of *Drosophila* also differentially express color-sensitive Rhs in the three butterfly ommatidial types, suggesting that the R7 signal also influences outer photoreceptor fate in butterflies. It is likely that honeybees use similar mechanisms for producing three ommatidial types (*Perry et al., 2016*). It will be interesting to see if the TGFβ pathways affect photoreceptor subtype specification in the visual systems of other insect species. Alternatively, residual pale cells could exist in *sev* mutants since removing signaling factors from R7 (*sev* mutant) would leave ligands in other photoreceptors intact, which may be permissive for a small amount of signaling. Indeed, we find expression of dActβ, Myo, Dpp and Amon in some outer photoreceptors. This may explain the difficulty in achieving fully penetrant phenotypes. In fact, Myo signals redundantly from astrocyte-like *and* larval cortex glia via Babo to remodel neurons of the mushroom body γ lobe (*Awasaki et al., 2011*). Further, dActβ signaling via Babo from both R7 *and* R8 cells is required in restricting arborizations of photoreceptor target neurons (*Ting et al., 2014*).

## Materials and methods

### Fly strains

The following transgenic and mutant *Drosophila* strains were used in this study:

UAS-babo* (**9B3**); dppRNAi (8A3); dppRNAi (8D2); dppRNAi (9B2-9B3); UAS-saxQD (HA1A); UAS-saxQD (HA3X); UAS-saxQD (HA5D3); tsgRNAi; sogRNAi; sogY121; UAS-babo-A; UAS-babo-B; UAS-babo-C; and sogY506 were generated in the M. O'Connor lab. Actβ$^{d80}$ (*Zhu et al., 2008*); UAS-Dcr2, ey-Gal4 + lGMR Gal4 (*Dietzl et al., 2007*); **melt:nucLacZ** reporter (*Mikeladze-Dvali et al., 2005*); **lGMR-Gal4** (*Wernet et al., 2003*); Rh5-GFP (*Pichaud and Desplan, 2001*); **sens-Gal4** (gift form G. Mardon); UAS-tkv$^{2ΔGSK}$ (*Haerry et al., 1998*); **wts-GFP** reporter (gift of J. Rister); **scabrous-Gal4** (*Hinz et al., 1994*); **prospero-Gal4** (*Ohshiro et al., 2000*); **PM1.81-Gal4** (*Lee et al., 2001*); **UAS-tkv*** (*Das et al., 1998*); **Dpp::GFP** (*Teleman and Cohen, 2000*).

ActβRNAi (**42493-RRID:BDSC_42493**; 29597-RRID:BDSC_29597; 42795-RRID:BDSC_42795); DppRNAi (**36779-RRID:BDSC_36779**; 25782-RRID:BDSC_25782; 31531-RRID:BDSC_31531; 33618-RRID:BDSC_33618); GbbRNAi (**34898-RRID:BDSC_34898**); DawRNAi (34974-RRID:BDSC_34974; 50911-RRID:BDSC_50911); MyoRNAi (**31200-RRID:BDSC_31200**; 31114-RRID:BDSC_31114; 36840-RRID:BDSC_36840); MavRNAi (34650-RRID:BDSC_34650; 36809-RRID:BDSC_36809); TkvRNAi (**41904-RRID:BDSC_41904**; 31040-RRID:BDSC_31040; 31041-RRID:BDSC_31041; 35166-RRID:BDSC_35166; 40937-RRID:BDSC_40937); UAS-Fur2 (**11429-RRID:BDSC_11429**); Fur2RNAi (25959-RRID:BDSC_25959; 51743-RRID:BDSC_51743); Fur1RNAi (25837-RRID:BDSC_25837; 41914-RRID:BDSC_41914); AmonRNAi (29009-RRID:BDSC_29009; **41635-RRID:BDSC_41635**; 29010-RRID:BDSC_29010; 44001-RRID:BDSC_44001); UAS-Sog (15500-RRID:BDSC_15500); SogRNAi (37405-RRID:BDSC_37405); wtsRNAi (**34064-RRID:BDSC_34064**); UAS-Yki (**28816-RRID:BDSC_28816**); Sca-lexA (**52763-RRID:BDSC_52763**); lexAOP-Gal80 (**32213-RRID:BDSC_32213**); UAS-Dcr2 (**24650-RRID:BDSC_24650**; **24651-RRID:BDSC_24651**); sev$^{14}$ (**5691-RRID:BDSC_5691**); PutRNAi (**35195-RRID:BDSC_35195**); WitRNAi (25949-RRID:BDSC_25949); Amon-Gal4 (**30554-RRID:BDSC_30554**); Babo-A-RNAi (44400-RRID:BDSC_44400); Babo-B-RNAi (44401-RRID:BDSC_44401); Babo-C-RNAi (44402-RRID:BDSC_44402) lines were obtained from the Bloomington Stock Center. baboRNAi (**106092**, 853); ActβRNAi (11062); ScwRNAi (**21400**, 21399, 105303); DawRNAi (105309, **13420**); TkvRNAi (105834, 3059); SaxRNAi (42436, **46350**, 46356, 9433, 9434); Fur2RNAi (**101242**); AmonRNAi (110788); TldRNAi (**1215**, 100930); SogRNAi (105853); TsgRNAi (14894, 45355, 108750); MeltRNAi (**105110**); MadRNAi (**12635**); dSmad2RNAi (**14609**); MedRNAi (19688, 19689); Myo reporter (**3180650**); dActβ reporter (**318136**) lines were obtained from the Vienna Drosophila Research Center.

Fly lines appearing in bold were used for experiments described in *Figures 2–6*. All other lines make up supplemental data.

Flies were raised on cornmeal–agar–molasses–yeast medium at 25°C for epistasis testing. y$^1$w$^{67}$; lGMR-Gal4, y$^1$w$^{67}$; UAS-Dcr2; lGMR-Gal4 and CantonS flies were used as wild-type controls for Rhodopsin expression.

## RNAi-based screen

The RNAi screen included 250 UAS-RNAi lines, which targeted around 100 membrane-associated factors. The Gal4 driver line contained both eyeless-Gal4 and lGMR-Gal4 drivers recombined on chromosome 2. Eyeless-Gal4 is expressed early in the entire eye disc, whereas lGMR-Gal4 is expressed only after the morphogenetic furrow and maintained in adults. Together, these two drivers induce RNAi expression in the whole eye from the time the eye is specified until adulthood. In addition, the RNAi driver stock carries UAS-Dicer2 to enhance the efficiency of generating small interfering RNA. Rh5-GFP, which is expressed only in pale R8 photoreceptors, was used as a readout in the screen. UAS-RNAi lines were crossed to the driver line (eyeless >Gal4, lGMR >Gal4; UAS-Dicer2; Rh5-GFP) at 25°C. The F1 progeny were analyzed under water immersion for a change in Rh5-GFP reporter expression (*Pichaud and Desplan, 2001*). Flies were raised at 29°C to strongly drive UAS-constructs for secondary screening purposes. Here, each line was dissected, immunostained and imaged according to methods outlined below.

## Sample size and replicates

Due to the robust nature of Rh5/Rh6 ratios in unique genotypic backgrounds, we looked at $\geq$5 biological replicates (never analyzing two retinas from one individual). This was sufficient to achieve extremely low variance in photoreceptor subtype ratios and little standard deviation from the mean. Experiments resulting in higher variance were technically replicated (indicated by N values of $\geq$10) to verify initial results.

## Immunohistochemistry, microscopy and quantification

Whole-mounted retina stainings were performed as previously described (*Cook et al., 2003*).

Primary antibodies used were: chicken anti-βGal (1:200, Gallus Immunotech, ABGTD, Canada), sheep anti-GFP (1:500, Bio-Rad, Hercules, California, 4745–1051), Alexa Fluor 488-conjugated Phalloidin (1:200, Molecular Probes, Eugene, Oregon), mouse anti-Prospero (RRID:AB_528440) (1:10, DSHB, Iowa City, Iowa), monoclonal mouse anti-Rh5 1:400 (RRID:AB_2567309) (*Chou et al., 1996*), guinea pig anti-Ss 1:100 (*Johnston and Desplan, 2014*), rabbit anti-Defective Proventriculus (Dve, RRID:AB_2568983) 1:250 (*Johnston et al., 2011*), rat anti-Rh6 1:1000; guinea pig anti-Rh4 1:500 (generated in this study). Secondary antibodies used were: Alexa Fluor 488-coupled made in donkey, anti-sheep, -guinea pig, -rabbit; Alexa Fluor 555-coupled made in donkey, anti-rat, -mouse, -guinea pig; Alexa Fluor 647-coupled made in donkey, anti-mouse, -chicken, -sheep (Molecular Probes). All secondary antibodies were used at a 1:800 dilution. Samples were mounted in Slowfade gold (Invitrogen, Carlsbad, California). Confocal microscopy was performed using a Leica SP5 confocal laser scanning microscope and processed with Leica AF-Lite software.

Peptide epitopes used for antibody production are as follows:

DmRh6: LACGKDDLTSDSRTQ

DmRh4: LGVNEKSGEISSAQS

The number of R8 cells that expressed Rh5, Rh6 or both were counted using custom-written Python code. R8 cells were identified by detecting local maxima in the Rh5 and Rh6 channels within a 10-pixel neighborhood. Local maxima with fluorescent intensities above an arbitrary threshold in either the Rh5 or Rh6 channel were counted as either Rh5- or Rh6-expressing cells; maxima with intensities above the threshold in both channels were counted as expressing both Rh5 and Rh6. The quality of the automatic labeling was assessed by eye for each sample. Custom Python source code is attached as *Source code 1*. Amon-expressing ommatidia were scored by hand.

## Acknowledgements

We would like to thank M O'Connor for reagents and helpful discussion; J Rister, S Britt, G Mardon, K Wharton, J Campos-Ortega, F Matsuzaki, S Zipursky, R Padgett, E Bach, the Bloomington Stock Center and the Vienna Drosophila Research Center for reagents; R Behnia, D Vasiliauskas, M Perry, J Rister and D Jukam for very helpful comments on the manuscript; Cleo Tsanis and Terry Blackman for technical support; the Desplan lab and Flynet for constant support.

## Additional information

### Funding

| Funder | Grant reference number | Author |
|---|---|---|
| National Eye Institute | EY022843-01 | Brent S Wells |
| National Eye Institute | EY13012 | Claude Desplan |
| European Molecular Biology Organization | ALTF 506-2002 | Daniela Pistillo |

The funders had no role in study design, data collection and interpretation, or the decision to submit the work for publication.

### Author contributions

Brent S Wells, Conceptualization, Data curation, Formal analysis, Supervision, Funding acquisition, Validation, Investigation, Visualization, Methodology, Writing—original draft, Project administration, Writing—review and editing; Daniela Pistillo, Conceptualization, Funding acquisition, Investigation; Erin Barnhart, Software, Writing—review and editing; Claude Desplan, Conceptualization, Formal analysis, Supervision, Funding acquisition, Project administration, Writing—review and editing

### Author ORCIDs

Brent S Wells http://orcid.org/0000-0003-1194-3795
Claude Desplan http://orcid.org/0000-0002-6914-1413

### Decision letter and Author response

Decision letter https://doi.org/10.7554/eLife.25301.017
Author response https://doi.org/10.7554/eLife.25301.018

## Additional files

### Supplementary files

• Source code 1. Custom Python source code.
DOI: https://doi.org/10.7554/eLife.25301.011

• Source data 1. Source data for all figures R8 photoreceptor subtype counts for each scored retina for all genotypes are listed with calculated average percentages of Rh5, Rh6, and Rh5 + Rh6 co-expressing cells. Standard deviation and standard error are also calculated for all groups. Genotype averages with standard error are presented and color coded in columns A-G (blue = Rh5, purple = Rh5 + Rh6 co-expression, red = Rh6).
DOI: https://doi.org/10.7554/eLife.25301.012

• Transparent reporting form
DOI: https://doi.org/10.7554/eLife.25301.013

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
