## [Decision Letter]

Thank you for submitting your article "Parallel Activin and BMP signaling coordinates R7/R8 photoreceptor subtype pairing in the stochastic *Drosophila* retina" for consideration by *eLife*. Your article has been reviewed by two peer reviewers and the evaluation has been overseen by a Reviewing Editor and Diethard Tautz as the Senior Editor. The following individuals involved in review of your submission have agreed to reveal their identity: Michael O'Connor (Reviewer #1); Lee Tzumin (Reviewer #2).

The reviewers have discussed the reviews with one another and the Reviewing Editor has drafted this decision to help you prepare a revised submission.

Summary:

A well-studied example of coordination of cell fate in the service of physiological function is the pairing of the R7 and R8 photoreceptors in the *Drosophila* retina. Randomly fated R7 cells confer the correct matching fate (Rh expression) to its paired R8 cell by setting the state of a bistable switch. The nature of the signal(s) between R7 and R8 is unknown. Here, expanding on a lead from an RNAi screen for altered R8 specification, the authors show that the TGFβ superfamily is required for pale R7+R8 ommatidia, and argue that multiple ligands form this signaling family act as the "signal" between R7 and R8. Notably, both Activin- and BMP-dependent TGF-β signaling pathways are involved with minimal redundancy. This finding is very interesting and the main conclusions are well documented.

Essential revisions:

Both reviewers commented that the primary issue is how the multitude of TGF-β ligands implicated are expressed, and how they would implement the binary cell fate decision. The reviewers would like to know which cell types express the various ligands, and also requested further data confirming the accuracy of the RNAi and mutant results for the ligands. Some details from their comments follow below. In this case, a satisfactory revision should contain additional data and discussion pertaining to the question of the TGF-β ligand activities.

1) "Although it is a bit buried in the literature, neither daw nor myo have been shown to be expressed in neurons (at least in larval brains) but both are expressed in glia. There are promoter/enhancer gal4 lines available for all three ligands and these should be checked for expression in the adult retina to determine if the myo and daw signals come from the R7 or glia. If neither of these ligands is really expressed in Rh7 or 8, then the RNAi results are quite questionable. Likewise there is available a dpp GFP line that can be used to examine the source of dpp."

2) "One should confirm the Spineless-negative R7s as the site of production for all, or just some, of the required TGF-β ligands. This is critical to substantiate TGF-β signaling as the instructive signal from R7. It could be done easily via targeted RNAi using the mentioned R7-specific drivers."

3) "The source of the ligands is relevant for two other issues. The authors suggest that heterodimers may be may account for some odd results, especially for myo and act β. However heterodimer formation requires expression in the same cell. So again determining which cells are the source of which ligands can identify which of several possible heterodimers might be relevant. The ligand source issue is also relevant to the authors’ final discussion point concerning how short range interactions are achieved."

---

## [Author Response]

Essential revisions:Both reviewers commented that the primary issue is how the multitude of TGF-β ligands implicated are expressed, and how they would implement the binary cell fate decision. The reviewers would like to know which cell types express the various ligands, and also requested further data confirming the accuracy of the RNAi and mutant results for the ligands. Some details from their comments follow below. In this case, a satisfactory revision should contain additional data and discussion pertaining to the question of the TGF-β ligand activities.1) "Although it is a bit buried in the literature, neither daw nor myo have been shown to be expressed in neurons (at least in larval brains) but both are expressed in glia. There are promoter/enhancer gal4 lines available for all three ligands and these should be checked for expression in the adult retina to determine if the myo and daw signals come from the R7 or glia. If neither of these ligands is really expressed in Rh7 or 8, then the RNAi results are quite questionable. Likewise there is available a dpp GFP line that can be used to examine the source of dpp."

We have checked most of the ligands using available reporter constructs. We first examined a dActβ protein reporter. Consistent with our prediction, dActβ was present in all R7 cells around 45-55hrs APF, which is when R8 subtype specification is thought to occur. It was also present in many R3 cells (Author response image 1).

**Author response image 1. respfig1:** (A) dActβ co-localized with the R7 marker Prospero (Pros) (circled in purple). It also weakly labeled some R3 cells. (A’) Pros marks R7 cells. (A”) dActβ was expressed in R7 and R3.

In addition to dActβ, a Myo protein reporter was also expressed at the same developmental time in all R7 cells as well as in R4 cells. A transcriptional reporter for Dpp was expressed in all PRs (see manuscript). We could not detect expression from a transcriptional reporter for Daw, possibly due to the molecular nature of the Gal4 insertion obscuring a necessary enhancer for retina expression. We were also unable to obtain cellular localization data for a Gbb reporter. Of note, the reporters that did work were all expressed in the R7 cell where we predict dimerization is necessary for signaling to the underlying R8 cell. These results support our hypothesis of an instructive signal for pale fate coming from R7 cells and not from outer PRs.

2) "One should confirm the Spineless-negative R7s as the site of production for all, or just some, of the required TGF-β ligands. This is critical to substantiate TGF-β signaling as the instructive signal from R7. It could be done easily via targeted RNAi using the mentioned R7-specific drivers."

To determine whether ligand expression was specific to a subtype of ommatidia, we co-stained with Spineless (Ss), which marks only yellow R7 cells. We found that dActβ was expressed in both pale and yellow R7s.

The other ligands, Myo and Dpp, were also expressed in all R7 cells, both pale and yellow. Therefore, we investigated whether a processing factor of TGFβ ligands was specific to pale cells, resulting in activation of downstream components specifically in the pale subtype.

To examine this, we tested a transcriptional reporter for the processing factor Amon, which gave an increased yellow phenotype when removed using RNAi. We found that Amon was expressed in a subset of R7 PRs. Strikingly, when Amon was expressed in R7 cells, it was almost exclusively in pale R7 cells (Author response image 2). We found that the expression of Amon was temporally regulated and was expressed in up to 33% of pale R7s, then tapered off drastically as the retina reached approximately 55hrs APF. However, even across an expanded temporal window, Amon was expressed in an average of 9% of pale R7 cells while it was only observed in less than 1% of yellow R7 cells (N = >2600 ommatidia). Therefore, Amon was expressed for a short period of time when R8 specification occurs with a very strong bias for pale R7 cells.

**Author response image 2. respfig2:** (A) Amon was expressed in pale R7 cells (circled in purple) but not in yellow R7s (circled in red). It was also expressed in some outer PRs (circled in blue) and absent from other pale R7 cells (circled in white). (A’) Ss marks yellow R7 cells (circled in red). (A”) Amon was present in some but not all Ss-negative pale R7s (circled in purple) but absent from Ss-positive yellow R7s (circled in red).

Interestingly, we found that Amon expression in R7 varies greatly across pupal retina, suggesting that its processing activity may be temporally activated and regionally transient (Author response image 3). This would explain the fact that Amon was never seen in all pale R7s and varied greatly across retinas and developmental age.

**Author response image 3. respfig3:** Across a single pupal retina, Amon expression varied, including in R7 cells. Dotted blue box represents 23% co-expression with Pros in R7, while the dotted white box represents 4% co-expression in another part of the retina.

We also set out to test Fur2 and Tld reporters. A Fur2 transcriptional reporter showed no expression, possibly due to timing or lack of necessary enhancer regions for expression in the retina. We were unable to obtain and investigate a Tld reporter in a reasonable timeframe. It remains possible that either of these processing factors is expressed in pale-specific R7 PRs.

3) "The source of the ligands is relevant for two other issues. The authors suggest that heterodimers may be may account for some odd results, especially for myo and act β. However heterodimer formation requires expression in the same cell. So again determining which cells are the source of which ligands can identify which of several possible heterodimers might be relevant. The ligand source issue is also relevant to the authors’ final discussion point concerning how short range interactions are achieved."

Please see above where we show that these genes are expressed in R7 cells.